# Wilms Tumor 1-Driven Fibroblast Activation and Subpleural Thickening in Idiopathic Pulmonary Fibrosis

**DOI:** 10.3390/ijms24032850

**Published:** 2023-02-02

**Authors:** Prathibha R. Gajjala, Priyanka Singh, Varshini Odayar, Harshavardhana H. Ediga, Francis X. McCormack, Satish K. Madala

**Affiliations:** Division of Pulmonary, Critical Care and Sleep Medicine, University of Cincinnati, Cincinnati, OH 45267-0564, USA

**Keywords:** idiopathic pulmonary fibrosis, fibroblast, collagen, lung function, extracellular matrix

## Abstract

Idiopathic pulmonary fibrosis (IPF) is a progressive fibrotic lung disease that is often fatal due to the formation of irreversible scar tissue in the distal areas of the lung. Although the pathological and radiological features of IPF lungs are well defined, the lack of insight into the fibrogenic role of fibroblasts that accumulate in distinct anatomical regions of the lungs is a critical knowledge gap. Fibrotic lesions have been shown to originate in the subpleural areas and extend into the lung parenchyma through processes of dysregulated fibroproliferation, migration, fibroblast-to-myofibroblast transformation, and extracellular matrix production. Identifying the molecular targets underlying subpleural thickening at the early and late stages of fibrosis could facilitate the development of new therapies to attenuate fibroblast activation and improve the survival of patients with IPF. Here, we discuss the key cellular and molecular events that contribute to (myo)fibroblast activation and subpleural thickening in IPF. In particular, we highlight the transcriptional programs involved in mesothelial to mesenchymal transformation and fibroblast dysfunction that can be targeted to alter the course of the progressive expansion of fibrotic lesions in the distal areas of IPF lungs.

## 1. Introduction

Pulmonary fibrosis is a pathological endpoint in many chronic lung diseases and is associated with repetitive lung injury, involving mesenchymal cell dysfunction and unremitting collagen deposition [1,2,3]. A key event in the manifestation of unresolved fibrosis is the persistent activation of fibroblasts, which culminates in myofibroblast accumulation and the excessive production of collagen and another extracellular matrix (ECM) proteins in the pulmonary parenchyma [4,5]. Pulmonary fibrosis is a major cause of death, as the progressive distortion of alveolar architecture impairs gas exchange [6,7]. Pulmonary fibrosis plays a major role in disrupting lung function in several chronic lung diseases, including idiopathic pulmonary fibrosis (IPF) and systemic sclerosis [4,8,9,10]. The activation of fibroblasts and collagen deposition are also implicated in the pathological progression of multiple lung cancers, resulting in the increased invasion and metastasis of oncogenic cells in tumors [11,12]. Therefore, the development of effective therapeutics against pulmonary fibrosis is an urgent pursuit in diverse research areas.

IPF is a chronic lung disease of unknown etiology with progressive scarring of the lungs and one of the most common forms of interstitial lung disease (ILD). Mortality and morbidity are increasing worldwide, with rates that are substantially higher in older populations (over 65 years of age), especially men [13,14]. The incidence of IPF is approximately 2.8–9.3 in 100,000 per year, and the median survival after diagnosis is approximately 3–5 years [15,16]. Pirfenidone and nintedanib are two recent U.S. Food and Drug Administration (FDA)-approved drugs that delay the decline in lung function but appear to have limited impact on the survival of patients with IPF [17,18]. The failure to develop more effective treatments is due in part to a lack of knowledge regarding the molecular mechanisms of disease pathogenesis, including factors that induce and sustain fibroblast activation. In this review, we discuss the pathogenesis of IPF, focusing mainly on the initiation of fibrotic lesions in the distal regions of the lung, mesothelial-myofibroblast transformation (MMT), and the current progress in identifying molecular nodes that maintain fibrotic niches in IPF.

## 2. Histological Features of IPF

The histological manifestation of IPF is usual interstitial pneumonia (UIP), with three main diagnostic features: (i) a patchwork pattern defined as a heterogeneous alternation of normal and scarred lung areas, (ii) the presence of a honeycomb pattern located in the subpleural parenchyma, and (iii) the presence of fibroblast foci [19]. The temporal and spatial heterogeneity of the lesions in IPF is characterized by the close proximity and sharp transitions between normal lung and fibrotic lungs (Figure 1). Fibroblastic foci serve as the leading edge of fibrosis, and the honeycomb pattern is indicative of the late stages of pulmonary fibrosis [20,21]. The subpleural localization of scarring in IPF is so characteristic that it is an integral part of making the diagnosis [22]. High-resolution computed tomography (HRCT) scanning demonstrating basilar predominant, subpleural reticular, and honeycomb patterns in patients with a compatible medical history can obviate the need for biopsy.

At the cellular level, the participation of fibroblasts in disease progression is well established and includes the aberrant activation of myofibroblasts, which are marked by smooth muscle actin alpha expression, that secrete excessive amounts of ECM proteins such as collagen and fibronectin [23,24]. Other hallmarks of fibroblast activation include the proliferation of fibroblasts, which is predominately limited to the early or expanding areas of fibrotic lesions in the lung parenchyma, excessive migration and invasiveness, fibroblast-to-myofibroblast transformation, and resistance to apoptosis [25,26,27,28,29]. In the following section, we discuss the potential role of mesothelial cells in myofibroblast activation, which is critical for the initiation and maintenance of subpleural fibrotic lesions in IPF.

## 3. Mesothelial Origin of Myofibroblasts

In IPF, myofibroblasts distort lung architecture by depositing excess ECM. The source of myofibroblasts is being investigated by lineage tracing in injury models for pulmonary fibrosis. Although several studies have implicated resident fibroblasts as the main precursors of myofibroblasts [30,31,32], other cell populations such as epithelial cells, fibrocytes [33,34], and pericytes [35] have also been reported to contribute to myofibroblast transformation and the expansion of fibrotic lung lesions [36]. In the past, epithelial cells were frequently cited as a major source of myofibroblasts; however, many studies have disproved the epithelial-mesenchymal transformation theory [26,37]. In particular, Hardie et al. [38] evaluated the contribution of epithelial cells for transforming the growth factor alpha (TGFα)-induced fibrosis in vivo. After labeling epithelial cells with β-galactosidase using a clara cell secretory protein (CCSP)/Cre driver, minimal to no staining was observed in the fibrotic lesions. Similar conclusions were drawn from other studies that used different epithelial-specific Cre drivers. Rock et al. [26] employed the surfactant protein C (Sftpc)-CreERT2 driver to label type 2 alveolar epithelial cells (ACE2) with a red fluorescent protein (RFP) in an intratracheal bleomycin model. They demonstrated that ACE2 cells do not contribute to fibroblasts that accumulate in fibrotic lung lesions. Similar conclusions were reached using the Secretoglobin Family 1A Member 1 (Scgb1a1)-CreER driver, which labeled clara cells as well as a few scgb1a1 and sftpc dual positive cells and concluded that epithelial cells were not the source of myofibroblasts in bleomycin-induced pulmonary fibrosis. However, epithelial-mesenchymal crosstalk plays a crucial role in activating fibroblasts and may enhance fibroblast-to-myofibroblast transformation (FMT) processes by secreting paracrine factors [39]. Understanding the impact of epithelial cells on FMT holds promise for improving IPF management.

Published studies have demonstrated an increase in bone-marrow-derived mesenchymal cells called fibrocytes both in circulation and fibrotic lung lesions associated with the progression of fibrotic lung remodeling in IPF [36,40]. This led us to question whether fibrocytes could contribute to the myofibroblast pool in pulmonary fibrosis, as well as their role in the progression of the disease. In the TGFα-mouse model, we were unable to demonstrate that transfused green fluorescent protein (GFP)-labeled fibrocytes contributed to the stroma of the fibrotic lung lesion [41]. Instead, the study provided evidence for the paracrine activation of resident lung fibroblasts by fibrocytes, supporting the notion of resident lung fibroblasts as the primary source of stromal cells [41]. Similar conclusions have been drawn from studies in renal fibrosis models, which suggest only a minor role in its contribution to the myofibroblast pool [42]. Likewise, it has been postulated that the pericyte, a type of mesenchymal cell that lines the capillaries and venules, may also contribute to the myofibroblast pool. Pericytes markers include neural/glial antigen 2 (NG2) and platelet-derived growth factor β (PDGFRβ). To test whether pericytes are the source of myofibroblasts in the bleomycin model, Rock et al. [26] utilized two mouse strains, Ng2-CreER and forkhead boxJ1 (FoxJ1)-CreER, to lineage-label pericyte-like cells. The lineage-labeled cells were proliferated in response to bleomycin; nevertheless, there was no evidence of colocalization with alpha-smooth muscle actin (αSMA), suggesting that pericytes were not a major contributor tp myofibroblasts in the fibrotic regions. In contrast, Hung et al. [30] utilized fate-mapping strategies and found that the Foxd1-expressing pericytes transform into myofibroblasts during bleomycin-induced injury. Although foxd1-derived pericytes transform into myofibroblasts, they are not the major source of myofibroblasts that accumulate during bleomycin-induced fibrosis. The differences in the observations made by Rock et al. [26] and Hung et al. [30] may be attributable to the differences in the labeling efficiency or to heterogeneity among pericyte cell populations.

A single sheet of cuboidal pleural mesothelial cells (PMCs) lines the lungs and expresses several epithelial and mesenchymal cell-specific genes, such as calretinin, cytokeratin, collagen, desmin, and vimentin, but not smooth muscle actin. Mesothelial cells can transform into myofibroblasts through the MMT process and may represent a novel source of myofibroblasts in the fibrotic lung [25,43,44,45,46]. Wilms tumor gene 1 (WT1) is a marker for mesothelial cells, and studies have shown that during embryonic development, the majority of lung resident fibroblasts are derived from the WT1-positive mesothelium [47] and populate the perivascular and peribronchial areas [48]. More recent studies have shown that certain tamoxifen-dependent Cre recombinase mouse models, such as CreERT2-driven recombination in Wilms tumor (WT1*^CreERT2^*) mice, are more reliable and reproducible than WT1^CreEGFP^ reporter mice [34,47,49,50]. The use of WT1*^CreERT2^* mice enabled the accurate labeling of WT1-positive mesothelial cells lining embryonic lungs, which were shown to ultimately give rise to mesenchymal cells of the lung parenchyma [47,49]. We demonstrated that WT1 is downregulated in the postnatal stages of lung development but is upregulated in mesothelial cells in IPF and in a mouse model of TGFα-induced pulmonary fibrosis [34]. Indeed, in vivo, postnatal mesothelial lung cells were transformed into myofibroblasts in TGFα/WT1^CreERT2/mTmG^ reporter mice during TGFα-induced pulmonary fibrosis. They were found in the subpleural areas of fibrotic lungs but not in the peribronchial or adventitial regions [32]. However, PMCs did not transform into myofibroblasts during single-dose bleomycin-induced injury (or adeno transforming growth factor beta1 (TGFβ1)-induced pulmonary fibrosis) [49], which might be because bleomycin-driven fibrosis is transient and lacks subpleural lesions that are similar to IPF. Recent studies using cultured PMCs have provided evidence for MMT in the pathogenesis of pulmonary fibrosis. In particular, the TGFβ1/SMAD3 axis has been implicated in MMT and myofibroblast accumulation in the parenchyma of TGFβ1-injured lungs [51]. Although these studies suggest that MMT contributes to subpleural fibrosis, molecular insights are limited, and the role of MMT in the initiation and expansion of fibrotic lesions in the distal airways and other areas of the lung is unclear [25,43,45,52]. Future studies are needed to elucidate both upstream and downstream WT1 targets and the possible crosstalk between the WT1-driven gene networks and the TGFβ/SMAD pathway in myofibroblasts. Understanding the complex regulation of myofibroblast formation by TGFβ-dependent and TGFβ-independent pathways in the pathogenesis of subpleural fibrosis in pulmonary fibrosis is essential for developing more efficacious therapeutics for IPF.

## 4. Molecular Insights on Fibroblast Dysfunction in IPF

Early abnormalities and the most rapid progression of IPF are predominantly observed in the subpleural regions, highlighting the need to understand the molecular mechanisms of subpleural fibrosis [34,53,54]. We have focused on a set of subpleural molecules, such as WT1 and Sox9, that play a prominent role in activating fibroblasts and promoting fibrotic events such as proliferation, migration, differentiation, and survival [32,55].

Many studies in the fibrosis field have identified integrin αvβ6 as a master regulator of pro-fibrotic processes that are produced primarily by injured epithelial cells and macrophages but also fibroblasts, myofibroblasts, and neutrophils [56,57]. TGFβ exerts SMAD-mediated actions on ECM production, inflammation, and myofibroblast formation: particularly the accumulation of apoptosis-resistant cells in IPF [58,59]. Nonetheless, emerging in vitro and in vivo evidence indicates that non-TGFβ/SMAD signaling pathways also contribute to myofibroblast transformation and pulmonary fibrosis [32,55]. In the following subsection, we review the emerging molecular targets of (myo)fibroblast activation in pulmonary fibrosis.

### 4.1. WT1

WT1 is a zinc finger transcription factor that plays a crucial role in the development of multiple organs, including the lungs, heart, and kidneys, and regulates post-transcriptional modifications and RNA metabolism [60]. Mutations or loss of WT1 in embryonic stages is associated with severe developmental defects and embryonic lethality in mice [60,61]. Expression levels of WT1 are low in adult mouse lung mesothelial cells, but it is upregulated in both mesothelial and mesenchymal cells in IPF lung tissue [32,34,62]. In our study, WT1 loss or gain-of-function studies in primary fibroblasts were sufficient to modulate fibroproliferation, myofibroblast formation, and ECM production [32]. Moreover, the genetic loss of WT1 markedly reduced the expression of ECM genes, such as collagen type1 alpha1 (*Col1α*) and collagen type V alpha 1 (*Col5α*), and proliferative genes, such as gremlin 1 (*Grem1*), runt-related transcription factor-1 (*Runx1*), wnt family member-4 (*Wnt4*), insulin-like growth factor 1 (*Igf1*), cyclin B1 (*Ccnb1*), and E2F transcription factor 8 (*E2f8*). Our cell fate mapping strategy, based on the lineage-specific expression of αSMA reporter fibroblasts, demonstrated that WT1 overexpression by transduction was sufficient to induce fibroblast to myofibroblast transformation (FMT). The motif analysis and chromatin immunoprecipitation experiments indicated that WT1 binds directly to the promoter DNA sequence of αSMA to induce the differentiation of FMT [32]. This revealed a sophisticated mechanism by which WT1 regulates FMT processes, highlighting the key role of WT1 in IPF. Previously, WT1 was shown to maintain the mesenchymal cell phenotype by repressing epithelial genes such as Snail (*Snail1*) and E-cadherin (*Cdh1*) during embryonic stem cell differentiation [63]. Notably, the haploinsufficiency of WT1 was sufficient to attenuate fibroproliferation, myofibroblast accumulation, and collagen deposition in both TGFα- and bleomycin-induced pulmonary fibrosis in vivo [32]. Our new findings suggest that WT1-driven effects on fibroproliferation are non-cell-autonomous and may involve paracrine factors secreted by WT1-expressing cells [32]. These results highlight the need for a more detailed investigation into the molecular mechanisms of WT1-driven fibroblast activation and pulmonary fibrosis and whether the crosstalk between WT1 and the TGFβ/SMAD pathway regulates them. Identifying WT1 as a positive regulator of fibroblast activation suggests a new target for treating fibrotic lung diseases and possibly for regulating fibrosis in other organs.

### 4.2. Aurora Kinase B

Aurora kinase B (AurkB) is a mitotic serine/threonine kinase involved in various stages of the cell cycle [64,65]. This molecule is highly expressed in different types of cancer and contributes to tumor progression through the increased proliferation and survival of the cells [65]. In the fibrotic field, for the first time, we have shown that AurkB is highly upregulated in fibroblasts of the subpleural region in IPF and in two alternative pulmonary fibrotic mouse models [66]. Its expression in IPF fibroblasts is regulated by WT1, as demonstrated by knockdown (KD) and the overexpression of WT1, and its binding to the AurkB promoter was validated by chromatin immunoprecipitation and promoter-driven luciferase assays. KD studies in both IPF and TGFα lung fibroblasts have demonstrated a pathogenic role for AurkB in fibrogenesis by promoting fibroproliferation and survival. Specifically, AurkB KD showed a marked reduction in proliferative genes such as cyclin A2 (*CCNA2*) and polo-like kinase (*Plk1*) and impacted the expression of pro-apoptotic genes such as *Bak*, *Bax*, and *Fas* in fibrotic fibroblasts. Furthermore, the inhibition of AurkB activity using barasertib in vitro resulted in altered fibroblast activation processes, such as proliferation and apoptosis. Treatment with barasertib in both bleomycin and TGFα fibrotic models rescued mice from fibrosis by attenuating collagen deposition and proliferation in vivo [66]. This study shows that the WT1-AurkB axis is a critical driver of fibroproliferation and survival. Therefore, targeting AurkB therapeutically with barasertib may highlight its potential benefits in IPF.

### 4.3. Heat Shock Protein 90

Heat shock protein 90 (HSP90) is an important molecule that has been extensively studied in organ fibrosis [67,68,69,70,71,72,73,74,75]. Its overexpression in subpleural compartments is implicated in the pathogenesis of pulmonary fibrosis, resulting in the regulation of key cellular processes apart from its chaperone activity [72]. HSP90AA and HSP90AB are the two isoforms of HSP90 that are well-studied in the context of fibrosis. They have common ATPase activity but also unique binding partners due to the lack of N-terminal signal peptides in HSP90AA. Under pathophysiological conditions, preferential binding to their partners allows them to perform different functions. Our laboratory and others have shown the pro-fibrotic activity of HSP90AB, which is able to regulate proliferation, ECM production, and myofibroblast transformation [72,76]. The KD of intracellular HSP90AB, but not HSP90AA, also attenuated pro-fibrotic genes such as *col1*α1, *col5*α1, and α*SMA*. However, both isoforms play important roles in fibroblast migration. Recently, Bellaye et al. [76] showed the synergistic role of HSP90AA and HSP90AB in myofibroblast transformation and survival. They demonstrated that HSP90AA was elevated in IPF, and its release into circulation was regulated by mechanical stress. The secreted HSP90AA signals via the lipoprotein receptor-related protein 1 (LRP1) and intracellular HSP90AB are essential to the stabilization of LRP1 and to amplify the HSP90AA-induced signal, thus regulating myofibroblast transformation. This indicates that both forms are pathogenic when expressed at higher levels than those under basal conditions. The authors also demonstrated that the ectopic treatment of fibroblasts with HSP90AA promotes αSMA expression independent of the TGFβ pathway, suggesting a spatio-temporal function of different isoforms. Currently, more than 10 HSP90 inhibitors that belong to multiple drug classes are in the advanced stages of clinical trials for cancer [77,78]. Most of these are small molecules that are derivatives of geldanamycin and block the activity of both isoforms. 17-N-allylamino-17-demethoxygeldanamycin (17-AAG) and 17-demethoxy-17-[[2-(dimethylamino) ethyl] amino]-geldanamycin (17-DMAG) bind to the ATP-binding pocket and change the conformation of the protein, leading to proteasomal degradation. In our study, we treated fibroblasts with 17-AAG to block the intracellular HSP90AA and HSP90AB forms, which attenuated fibroblast activation and TGFβ-induced myofibroblast transformation. Moreover, the pharmacological inhibition of HSP90 with 17-AAG or 17-DMAG in a pulmonary fibrosis model has attenuated ongoing and established fibrosis, highlighting the potential benefits of HSP90 inhibition in IPF [72,79]. In a study by Bellaye et al. [76], HS-30, a non-permeable HSP90 inhibitor, was used to target the extracellular HSP90AA in precision-cut lung slices. The authors demonstrated the effects by inhibiting the extracellular HSP90 AA form, suggesting the unique features of different isoforms. However, characterization of the extracellular HSP90AA inhibitory effects in the pulmonary fibrosis models is necessary to shed light on how HSP90 functions. Nevertheless, the emergence of a growing body of evidence suggests that HSP90 is an important target with the potential for future therapies in pulmonary fibrosis.

### 4.4. Sox9

Sox9 belongs to the SOX family of proteins that are characterized by the highly conserved high mobility group (HMG) domain of sex-determining region Y (Sry) proteins [80]. Sox9 is selectively expressed by epithelial progenitor cells to modulate branching morphogenesis in the lung and the organized deposition of collagen as a part of cartilage formation in multiple organs, melanocyte differentiation, and male gonad development [81,82,83,84,85]. The dysregulation of Sox9 has been shown to be associated with the development of different types of cancer [86] and fibrosis in multiple organs, including the lung, kidney, heart, and liver [55,87,88,89]. Our recent findings showed the aberrant Sox9 overexpression in fibroblasts that accumulate in the subpleural, peribronchial, and fibrotic foci of IPF lungs [55]. This was further validated by the upregulation of Sox9 in distal lung fibroblasts derived from IPF lungs and in TGFα-overexpressing mice with severe fibrotic lung disease. The promoter-driven luciferase assay suggests the direct binding of WT1 to the *Sox9* promoter in the presence of TGFα, which, consistent with the upregulation of Sox9 in the lung fibroblasts of IPF patients, is positively regulated by the TGFα-WT1 axis. The loss of Sox9 in IPF fibroblasts is sufficient to attenuate the expression of fibrosis-associated genes such as ECM genes and genes associated with mesenchymal cell differentiation and growth. Similarly, the overexpression of Sox9 in fibroblasts resulted in the upregulation of pro-fibrotic growth factors such as TGFβ1, IL-6, IL-13, and IL-17, but the mechanisms underlying Sox9-driven fibrosis in the early and late stages of fibrosis are yet to be determined. Hence, studying Sox9-driven molecular networks and signaling pathways is a promising approach for identifying potential therapeutic candidates for IPF and other fibrotic diseases. A recent study by Jiang et al. [90] demonstrated that the vascular endothelial growth factor (VEGF) receptor 2 (kinase insert domain receptor (KDR)) loss mediated Sox9 overexpression in airway mucous metaplasia in asthma and cystic fibrosis (CF) patients. These new findings further support the potential role of Sox9 in the pathogenesis of other chronic lung diseases with dysregulated epithelia and mesenchyme.

### 4.5. Other Key Regulators of (Myo)fibroblast Activation in Pulmonary Fibrosis

Fox head box M1 (Foxm1) is a well-known cell cycle regulator that belongs to a family of transcription factors characterized by forkhead DNA binding domains. It acts downstream of the phosphoinositol-3-kinase (PI3K)-AKT signaling cascade. Penke et al. showed the upregulation of FOXM1 in fibroblasts isolated from the IPF lung [91]. The fibroblast-specific deletion of FOXM1 resulted in a reduced expression of several profibrotic genes such as αSMA, connective tissue growth factor (CTGF), Col1α1, and Tgfβ1. Fibroblast-specific Foxm1 deleted mice were also protected against bleomycin-induced fibrosis [91]. Recent studies have also demonstrated how FOXM1 suppression inhibits fibroblast differentiation to myofibroblasts during pulmonary fibrosis [92,93,94,95]. Another Fox protein called FOXL1 was found to be elevated in IPF lungs, potentially contributing to fibroblast accumulation in fibrotic lung lesions by activating TAZ (transcriptional coactivator with PDZ-binding motif) and YAP (Yes-associated protein) cascades and the PDGF axis via PDGFRα (platelet-derived growth factor receptor-α) [96].

Dock2 (Dedicator of cytokinesis 2) is an evolutionarily conserved guanine nucleotide exchange factor that activates Rac and regulates leukocyte migration and activation. Qian et al. reported elevated levels of Dock2 and colocalization with αSMA in the thickened pleura of nonspecific pleuritis patients [97]. The study also showed that the TGF-β is responsible for DOCK2 expression in human pleural mesothelial cells (PMCs) through meso MT processes. Furthermore, DOCK2 knockdown attenuated the expression of profibrotic genes such as αSMA, Col1A1, and fibronectin1. They also demonstrated that Tgfβ–induced MesoMT and Dock2 overexpression modulated Snail expression via Smad3 in PMCs [97]. In another study, elevated DOCK2 expression was observed in fibroblasts isolated from IPF and the bleomycin model [97]. The authors also showed that TGFβ–induced DOCK2 overexpression is dependent on both SMAD and ERK signaling.

Overall, the studies highlighted here suggest that a comprehensive understanding of both cellular and molecular mechanisms underlying fibrosis in the distal areas of the lung is critical for the development of new therapies against IPF. Fibroblasts and myofibroblasts are the primary targets to attenuate excessive ECM deposition in severe fibrotic lung diseases. These cells display significant heterogeneity, which is evidenced by the differential expression of markers such as Thy1 and differences in their lipid content, cytoskeletal composition, and cytokine profile. Multiple single-cell RNA sequencing (scRNA-seq) studies from both humans and mice have demonstrated morphologically and functionally distinct fibroblasts from IPF compared to normal lungs. The list of fibroblast populations includes myofibrogenic mesenchymal fibroblasts (Axin^+^), mesenchymal alveolar niche (Axin2^+^PDGFR^+^)^,^ fibroblasts (Lgr6^+^), fibroblasts involved in alveolar differentiation (Lgr5^+^), collagen-producing (Cthrc1^+^), profibrotic mesenchymal cells (PDGFRb^hi^) and pleural ECM-producing fibroblasts (Has1^hi^) [24,31,98,99,100]. Our recent studies using preclinical models and the immunostaining of IPF lungs have demonstrated the accumulation of myofibroblasts that express high levels of profibrotic transcription factors, including WT1 and Sox9, in the fibrotic lesions of IPF [32,34,55,99,101]. The accumulation of these profibrotic populations was further validated in recent scRNA-seq studies (Figure 2). Habermann et al. reported multiple fibroblast subtypes, including HAS1-positive fibroblasts, that expressed WT1 and were selectively accumulated in subpleural fibrotic lesions of IPF (Habermann et al., 2020a). Our studies showed the accumulation of Sox9-positive fibroblasts in both subpleural fibrotic lesions and the peri-bronchial fibrotic lesions of IPF lungs (Figure 2).

## Figures and Tables

**Figure 1 ijms-24-02850-f001:**
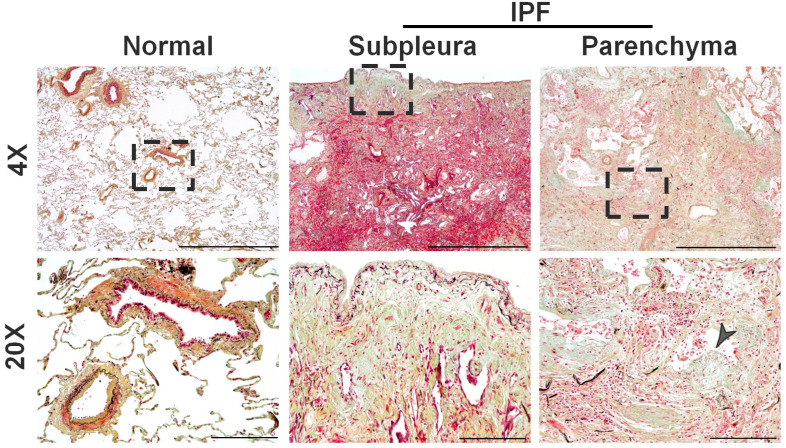
Representative images of Movat Pentachrome-stained distal areas of normal and IPF lungs. Highlighted dashed region in low magnification (Scale bar, 1500 µm) images represents the high magnification (Scale bar, 200 µm) images that show the prominent subpleural thickening and fibrotic foci that accumulate in the distal areas of the alveolar parenchyma of IPF lungs compared to normal lungs. Pentachrome staining highlights collagen (yellow color), muscle (red color), and elastic fibers (black to blue color) in mature fibrotic lesions of IPF. Arrowhead is used to highlight the fibrotic foci.

**Figure 2 ijms-24-02850-f002:**
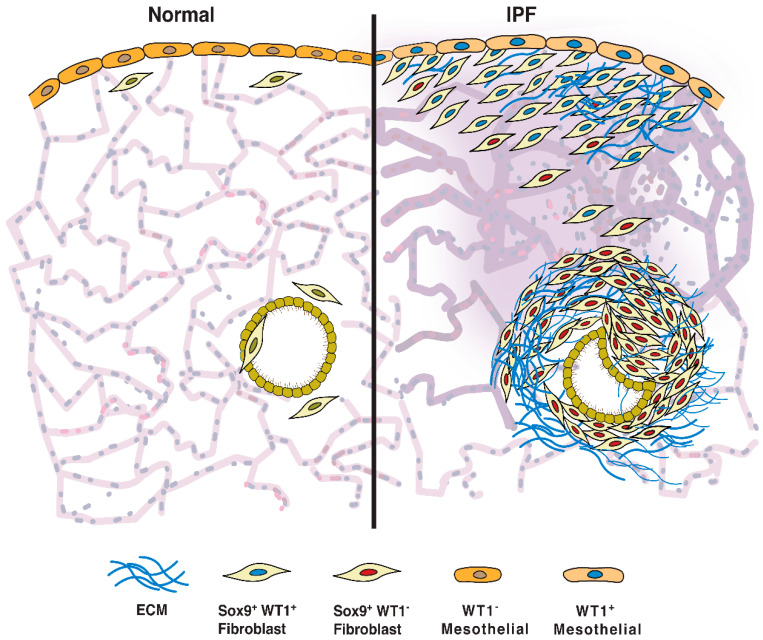
Accumulation of WT1- and Sox9-positive or Sox9-positive fibroblasts in IPF. WT1-mediated Sox9 upregulation in mesothelium and fibroblasts induces accumulation of fibroblasts that are positive for both WT1 and Sox9 in subpleural fibrotic lesions. Additionally, we observed accumulation of Sox9 positive fibroblasts that are negative for WT1 in peribronchial fibrotic lung lesions. Both WT1 and Sox9 expressing fibroblasts can transform to myofibroblasts resulting in progressive accumulation in fibrotic lesions of IPF lungs (**right**) compared to normal lungs (**left**).

## Data Availability

Not applicable.

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
