# Peer review of "Wilms Tumor 1-Driven Fibroblast Activation and Subpleural Thickening in Idiopathic Pulmonary Fibrosis"

_ijms, 2023, doi:10.3390/ijms24032850_

Round 1

Reviewer 1 Report

The review by Prathibha Gajjala and colleagues describes the importance of molecular events involved in the activation of fibroblasts to myofibroblasts and subpleural thickening leading to pathogenesis of progressive pulmonary fibrosis. The review focuses mainly on transcriptional program involved in mesothelial to mesenchymal transition and fibroblast dysfunction. Despite of extensive focus on the study of radiological and pathological features of IPF, the definitive mechanistic information that promotes fibroblasts activation and fibrogenesis is scanty. Therefore, the review is timely, interesting and discusses important mechanistic questions related to the pathogenesis of subpleural pulmonary fibrosis.  Having said all these good things about the current review, addressing the below listed review points may improve overall presentation and contribution. 

The authors should provide some background information about other pathways besides WT1, Aurora kinase, HSP90 and sox9 involved in activation of fibroblasts to broaden their contribution.

The significance of meso-mesenchymal transition (MMT) to the pathogenesis of progressive pulmonary fibrosis may be limited compared to other contributing mechanism, which needs to be acknowledged.

Author Response

  1. The authors should provide some background information about other pathways besides WT1, Aurora kinase, HSP90 and sox9 involved in activation of fibroblasts to broaden their contribution.

Response:  The manuscript has been revised and included a new section on other key molecules and pathways involved in fibroblast activation and pulmonary fibrosis (Pages, 9-10; Lanes, 328-335).

2. The significance of meso-mesenchymal transition (MMT) to the pathogenesis of progressive pulmonary fibrosis may be limited compared to other contributing mechanism, which needs to be acknowledged.

Response: We completely agree with the reviewer, and we now clearly stated the limitations of MMT in the progressive expansion of fibrotic lesions in other areas of the lung (Page 5; Lanes, 179-182).

Reviewer 2 Report

Idiopathic pulmonary fibrosis (IPF) is a progressive fibrotic complication that often leads to death. The radiological and pathological characteristics of this disease are very well defined. However, the lack of knowledge about the fibrogenic role of fibroblasts that accumulate in distinct anatomical regions of the lungs is a major impediment to the development of effective therapies against IPF.

This review elegantly discusses the different possible molecular and cellular events that may participate in fibroblast/myofibroblast activation and thickening of the subpleural region in idiopathic pulmonary fibrosis. The authors also report the transition programs that are involved in the famously important mesothelial to mesenchymal transformation and fibroblast dysfunction that can be targeted to alter the course of progressive expansion of fibrotic lesions in the distal areas of the lungs of IPF.

The review is very well presented and extremely pleasant to read.

Author Response

This review elegantly discusses the different possible molecular and cellular events that may participate in fibroblast/myofibroblast activation and thickening of the subpleural region in idiopathic pulmonary fibrosis. The authors also report the transition programs that are involved in the famously important mesothelial to mesenchymal transformation and fibroblast dysfunction that can be targeted to alter the course of progressive expansion of fibrotic lesions in the distal areas of the lungs of IPF. The review is very well presented and extremely pleasant to read.

Response: We are very grateful and appreciate the efforts put into the review of the manuscript. We believe that the revised version is a significant improvement from our initial submission by focusing with greater depth on other molecules involved in fibroblast activation and pulmonary fibrosis.

Reviewer 3 Report

Comments

1)    Abbreviation in title and subtitle must be avoided

2)    Title may be modified as it reflects that the review is specifically about the role of WT1.

3)    Introduction:

Clarification about WT1 and WT must be provided.

Detail information of fibroblast heterogeneity of IPF must be incorporated

4)    In section 2, The features of IPF such as highly elevated fibroblasts including excessive migration and invasiveness, fibroblast-to-myofibroblast transformation,  and resistance to apoptosis  etc must be incorporated along with references.

5)    “Sftpc”, “AurkB” has been written in different format in the manuscript. Please unify them.

6)    Diagnostic feature of IPF such as honeycomb pattern, fibroblast foci and subpleural regions, subpleural fibrosis etc must be shown and labelled in relevant fig.

7)    Line 165, it is written, “Early abnormalities ….. subpleural fibrosis”. Please provide references.

8)     Line 167, it is written  “We have focused on a set of  subpleural molecules .. .….. and survival”, here name of subpleural molecules must be provided.

Conclusion: Sufficient literature report of mesothelial to mesenchymal transformation in subpleural fibrosis must be provided.

Difference between IPF and other closely associated disease, protein responsible for the pathogenesis of IPF, protein profiling studies, subpleural molecules etc information could have been incorporated and analysed in order to draw appropriate conclusion.

A list of  key  cellular and molecular events that contribute to (myo) fibroblast activation and subpleural thickening in IPF could have been incorporated along with references.

At this stage the review is a few collective information without sufficient analysis and appropriate conclusion.

  Various abbreviation such as FDA, TGFa, CCSP/Cre, RFP, RT-PCR,GFP, PMC, FMT,KD, VEGF,KDR, CF etc. has been used. Full form of all abbreviation must be provided when appear 1st time in the manuscript.

Author Response

1)    Abbreviation in title and subtitle must be avoided

Response: We have replaced the abbreviation for WT1 with “Wilms Tumor 1” and modified the title to reflect the review points.

2)    Title may be modified as it reflects that the review is specifically about the role of WT1.

Response: As suggested the new title that reflects the review is now included “Wilms Tumor 1-driven fibroblast activation and subpleural thickening in idiopathic pulmonary fibrosis”

3)    Introduction:

Clarification about WT1 and WT must be provided. Detail information of fibroblast heterogeneity of IPF must be incorporated

Response: As suggested we now included additional details on fibroblast heterogeneity (Page 10-11; Lanes, 359-377) and also clarified WT1 versus WT.

4)    In section 2, The features of IPF such as highly elevated fibroblasts including excessive migration and invasiveness, fibroblast-to-myofibroblast transformation, and resistance to apoptosis, etc must be incorporated along with references.

Response: As suggested we now included appropriate references.

5)    “Sftpc”, “AurkB” has been written in different format in the manuscript. Please unify them.

Response: This has been addressed in the revised version of the manuscript.

6)    Diagnostic feature of IPF such as honeycomb pattern, fibroblast foci and subpleural regions, subpleural fibrosis etc must be shown and labelled in relevant fig.

Response: We included additional images to highlight subpleural thickening and fibrotic foci in the fibrotic lesions of IPF (Figure 1).

7)    Line 165, it is written, “Early abnormalities ….. subpleural fibrosis”. Please provide references.

Response: As suggested we now included appropriate references (Page 6; Lanes 192-195).

8)     Line 167, it is written  “We have focused on a set of  subpleural molecules .. .….. and survival”, here name of subpleural molecules must be provided.

Response: As suggested additional details are included to improve the clarity of the sentence on subpleural molecules (Page 6; Lanes, 193-194).

Conclusion:

  1. Sufficient literature report of mesothelial to mesenchymal transformation in subpleural fibrosis must be provided.

Response: This is an excellent suggestion, and we included additional references on MMT (Page 5; Lanes, 155-156). 

Difference between IPF and other closely associated disease, protein responsible for the pathogenesis of IPF, protein profiling studies, subpleural molecules, etc information could have been incorporated and analysed in order to draw appropriate conclusion. A list of  key  cellular and molecular events that contribute to (myo) fibroblast activation and subpleural thickening in IPF could have been incorporated along with references.

Response: As suggested we have included other key cellular and molecular events that contribute to fibroblast activation are now included (Pages, 9-10; Lanes, 328-335). We limited our discussion to subpleural fibrosis in IPF as limited insights were available with other pleural diseases.

At this stage the review is a few collective information without sufficient analysis and appropriate conclusion.

Response: We are very grateful to the reviewer for valuable suggestions, and we have included extensive changes in the manuscript to highlight the analysis of the literature and provided conclusions.

Various abbreviation such as FDA, TGFa, CCSP/Cre, RFP, RT-PCR,GFP, PMC, FMT,KD, VEGF,KDR, CF etc. has been used. Full form of all abbreviation must be provided when appear 1st time in the manuscript.

Response: We revised the manuscript with form of all abbreviations as needed.

Reviewer 4 Report

I believe it is an excellent review on an increasingly relevant topic, which by studying the pathogenesis and pathophysiology, helps us to better understand the mechanisms that condition this fibrotic damage that conditions greater morbidity and mortality in the unfortunate patients.

In the same way, expanding the study of the cell lines involved can help to find better therapeutic options, since they are currently very small and with low success rates.

I consider it is well organized and structured and should be published

Author Response

I believe it is an excellent review on an increasingly relevant topic, which by studying the pathogenesis and pathophysiology, helps us to better understand the mechanisms that condition this fibrotic damage that conditions greater morbidity and mortality in the unfortunate patients.

In the same way, expanding the study of the cell lines involved can help to find better therapeutic options, since they are currently very small and with low success rates.

I consider it is well organized and structured and should be published

Response: We like to thank the reviewer for valuable suggestions, and we feel that the revised version is a significant improvement from our initial submission by focusing with greater depth on subpleural fibrosis.

Round 2

Reviewer 3 Report

comments

1.Country name must be provided in the address of authors.

Introduction:

1.     As per response to comment 4, the new reference must have been mentioned in the response file. It is not clear that which reference has been incorporated and where.

Other comments

1.Manuscript must be carefully revised as there are mistakes such as the word “fibroproliferation” in line 101 and many more. It must be revised appropriately.

2. More relevant references must be incorporated in order to strengthen the review paper. 

Author Response

1.Country name must be provided in the address of authors.

Response: As suggested, we now included the city and country name (Page1, lanes 10-12)

Introduction:

  1. As per response to comment 4, the new reference must have been mentioned in the response file. It is not clear that which reference has been incorporated and where.

Response: Our sincere apologies. Suggested references are now added in section 2 on histological features of IPF (Page 3, lanes 92-93).

Other comments

  1. Manuscript must be carefully revised as there are mistakes such as the word “fibroproliferation” in line 101 and many more. It must be revised appropriately.

Response: We thank the reviewer for identifying minor edits needed to improve the readability of the text. We now thoroughly reviewed the manuscript for language edits and incorporated the corrections needed.

  1. More relevant references must be incorporated in order to strengthen the review paper. 

Response: We now included additional references needed to strengthen the review (Lanes 34-35, 40, 42-43, 45-46, 81, 104-106, and 158-159).

Round 3

Reviewer 3 Report

Minor spell check is required